# Prospective Randomized Controlled Clinical Trial to Evaluate the Safety and Efficacy of ACTLINK Plasma Treatment for Promoting Osseointegration and Bone Regeneration in Dental Implants

**DOI:** 10.3390/bioengineering11100980

**Published:** 2024-09-29

**Authors:** Jin-Seon Kwon, Won-Tak Cho, Jong-Ho Lee, Ji-Young Joo, Jae-Yeol Lee, Youbong Lim, Hyun-Jeong Jeon, Jung-Bo Huh

**Affiliations:** 1Department of Prosthodontics, School of Dentistry, Pusan National University, Yangsan 50612, Republic of Korea; cindy567@naver.com; 2Department of Prosthodontics, Dental Research Institute, Dental and Life Sciences Institute, Education and Research Team for Life Science on Dentistry, School of Dentistry, Pusan National University, Yangsan 50612, Republic of Korea; joonetak@hanmail.net; 3Research and Development Institute, PNUADD Co., Ltd., Busan 46241, Republic of Korea; addlee85@naver.com; 4Department of Periodontology, School of Dentistry, Pusan National University, Yangsan 50612, Republic of Korea; joojy@pusan.ac.kr; 5Department of Oral and Maxillofacial Surgery, School of Dentistry, Pusan National University, Yangsan 50612, Republic of Korea; id9753153@gmail.com; 6Plasmapp Co., Ltd., Yongin-si 17086, Republic of Korea; ceo@plasmapp.com (Y.L.); hjjeon@plasmapp.com (H.-J.J.)

**Keywords:** plasma treatment, implant surface modification, osseointegration, marginal bone change

## Abstract

Recent studies have explored surface treatments, such as increasing the hydrophilicity of implant fixtures, to enhance the osseointegration of implants. This prospective clinical study aimed to assess the clinical stability and efficacy of plasma treatment applied to implants with sandblast−acid etching (SLA) surfaces before placement. Twenty-eight patients requiring implant placement provided consent and were assigned randomly to either the SLA group without plasma treatment or the SLA/plasma group with plasma treatment. Recall checks were conducted one and three months after the first-stage surgery, followed by a second surgery at four months. Although no significant differences in buccal bone defects or implant stability were observed between the groups, the SLA/plasma group showed significant increases in marginal bone changes on the mesial and distal sides, as assessed using periapical radiographs. This study underscores the potential of pre-implantation plasma treatment to enhance bone regeneration around implants.

## 1. Introduction

Osseointegration, introduced by Per−Ingvar Brånemark, refers to the direct structural and functional connection between the surface of the dental implant and the living bone [1]. Research on the surface characteristics to enhance the osseointegration of implants is ongoing [2]. The surface treatment of fixtures through sandblasting and acid etching (SLA) is common practice [3]. On the other hand, such titanium surfaces undergo biological aging, meaning that their hydrophilicity and biological properties diminish over time [4,5,6]. Therefore, there has been research on methods such as ultraviolet (UV) [7] or plasma treatment [8] to enhance the hydrophilicity of implant fixtures and remove carbohydrate contamination from the implant surface, thereby promoting the key processes of osseointegration. 

In a study comparing the effects of UV−A and UV−C treatment on a titanium implant surface for 40 min, the results showed that the contact angle of untreated implants was over 90°, which decreased to below 5° after UV−A treatment and to 34° after UV−C treatment [7]. A study comparing cell adhesion and mRNA expression in vitro after UV and plasma treatment reported that a 12-minute UV treatment and plasma treatment for 1 minute resulted in the highest cell growth [9]. In addition, the research findings have suggested no significant differences in implant stability and marginal bone loss after UV−C treatment in vivo [10]. 

Plasma is an ionized gas in which electrons and ions move freely, and it can be used in various fields [11,12,13,14]. It is typically formed when high temperatures or electrical energy are applied, consisting of ions, electrons, neutral particles, and various forms of radiation, such as UV. In a literature review study on surface modification using plasma, it was noted that plasma can be effectively used to create antibacterial coatings, and ongoing research aims to induce antibacterial and osteoinductive properties in metallic biomaterials through plasma treatment. Additionally, it was stated that plasma has the potential to meet biocompatibility, cost-effectiveness, reproducibility, and industrial productivity requirements for the surface treatment of biomaterials [15]. In another review article on surface treatment using plasma, it was reported that plasma exhibits properties that enhance surface adhesion, corrosion resistance, hardness, and wear resistance when applied to metal surfaces [16]. Additionally, plasma can enhance the biocompatibility of the surface while retaining the favorable bulk material properties of biomaterials, which plays a crucial role in their integration with the biological environment [17].

Plasma can be categorized into thermal and non-thermal types [17], and non-thermal plasma can be generated at room temperature and has the advantage of short processing times. Previous studies analyzing surface characteristics after plasma treatment in vitro have reported increased hydrophilicity and surface energy, facilitating extensive cell attachment to the implant surface [18,19,20]. Transforming hydrophobic implant surfaces into hydrophilic ones may enhance the compatibility with blood, blood proteins and cells, and bone tissue around the implantation site [21]. Moreover, in vitro studies on the maintenance period of surface characteristics after plasma surface treatment have reported a 30-day duration [22]. In animal studies, implants treated with chairside non-thermal vacuum plasma treatment showed increased dental implant osseointegration, as evidenced by reduced marginal bone loss, and this study evaluated the potential for shorter healing times [23]. Therefore, a prospective clinical trial was conducted to assess the safety and efficacy of plasma treatment on fixtures in patients requiring implant placement.

## 2. Materials and Methods

### 2.1. Randomized Controlled Clinical Trial

This study was a prospective, single-blinded, non-inferiority, randomized, and controlled clinical trial to evaluate the safety and efficacy of vacuum plasma treatment using the ACTILINK System (Plasmapp Co., Ltd., Daejeon, Republic of Korea). According to the most recent study by Jung et al. [24] evaluating the efficacy of bone regeneration surgery using implants, the new bone regeneration rate was the highest, with a mean ± standard deviation of 96.4 ± 10 (%). We assume the lowest mean value from previous studies, i.e., 86% from Hämmerle et al. [25], as the “acceptable maximum value” for satisfying the non-inferiority of this study. For sample size calculation, a one-sided significance level of 0.025 and a power of 80% were assumed. The number of subjects needed to satisfy the non-inferiority hypothesis was calculated using PASS 13 (NCSS Statistics Software, Kaysville, UT, USA), resulting in 14 subjects per group, for a total of 28 subjects included in this clinical study. This prospective study protocol was approved by the Institutional Review Board of Pusan National University Dental Hospital (IRB Approval No. 2023-05-020).

### 2.2. Plasma Treatment of the Dental Implant

The ACTILINK system is a device developed to reduce surface impurities of the dental implant by vacuum plasma treatment (Figure 1). For plasma treatment, the dental implant is connected to the fixture driver. The fixture driver used here is one recommended by the manufacturer of the dental implant. The fixture driver with the dental implant is mounted on a dedicated holder for the ACTILINK system. When the prepared holder is mounted on the ACTILINK system, the implant is electrically grounded to the device. The set vacuum plasma process is performed by pressing the button. When the process begins, the tube of ACTILINK system is lowered to the base and the inflow of external air is blocked. After that, a vacuum of less than 10 torr is formed inside the tube through pumping, and a voltage of approximately 3 kV is supplied to the implant from the power electrode at the top of the tube to discharge plasma on the surface of the implant. Impurities on the implant surface are removed through plasma treatment, and after plasma treatment, impurities are removed through additional pumping using a vacuum pump. This process takes about 1 min.

### 2.3. Patients and Case Selection

Twenty−eight patients requiring dental implant treatment were recruited consecutively at Pusan National University Dental Hospital in the Republic of Korea. Informed consent was obtained from all the patients prior to the study’s initiation. From July 2023 to January 2024, 28 patients who underwent implant insertion at the Department of Periodontology, Oral and Maxillofacial Surgery and the Department of Prosthodontics at Pusan National University Dental Hospital were included. The subjects who met the inclusion and exclusion criteria received clinical and radiographic screening to confirm the residual bone quantity (Table 1) and were experienced in implant surgery with guided bone regeneration (GBR).

### 2.4. Treatment Group Allocation

Twenty-eight participants were randomly assigned into two groups using Microsoft Excel 2016 (Microsoft Corp., Redmond, WA, USA). A list of participants was entered, and random numbers were generated to allocate participants to groups randomly based on these numbers.
SLA group: non-plasma-treated SLA implants, n = 14.SLA/plasma group: plasma-treated SLA implants, n = 14.

### 2.5. First-Stage Surgical Procedure and Plasma Treatment on Implants

The first-stage surgery for implant placement and GBR was performed under local anesthesia. A mid-crestal incision and, if necessary, a vertical release incision were made on the edentulous ridge, and the mucoperiosteal flap was dissected and elevated. The SLA implant (Addpalnt On Implant System, PNUADD, Busan, Republic of Korea) was inserted in a prosthetically ideal position, and the initial stability of the implant was assessed immediately using an Ostell^®^ mentor device (Integration Diagnostics AB, Göteborg, Sweden) and Easy Check (Dentium, Seoul, Gyeonggi-do, Republic of Korea). For the SLA/plasma group, dental implant fixtures underwent plasma treatment using the ACTILINK System device before implant insertion. The plasma-treated implant fixtures were implanted in patients immediately after the treatment. The peri-implant dehiscence defect height and depth were measured using a periodontal probe (Colorvue Tip, Hu-Friedy, Leimen, Germany) (Figure 2) and augmented with bovine bone mineral (BONE−D XB, Medpark, Busan, Republic of Korea). The augmented defect was covered with collagen membrane (Colla−DM, MedPark, Busan, Republic of Korea), and the surgical site was closed using absorbable sutures (4−0 Vicryl^®^, Ethicon, Somerville, NJ, USA). All implants were in a submerged position with a cover screw for four months. The subjects were instructed to refrain from mechanical oral hygiene in the surgical site and to use a 0.12% chlorhexidine mouth rinse regimen (Hexamedine, Bukwang, Seoul, Republic of Korea) and prescribed antibiotics (Augmentin^®^ 625 mg, amoxicillin/clavulanic acid) and analgesics (Tylenol^®^, acetaminophen). The suture was removed 7–10 days after first-stage surgery.

### 2.6. Second-Stage Surgical Procedure

The second-stage implant surgery was performed four months after implant placement. Similarly to the first-stage surgery, a mid-crestal incision was made in the edentulous ridge, and vertical releasing incisions were performed when necessary. The mucoperiosteal flap was dissected and elevated. The bone defect height and width were measured using the same method as the first-stage surgery. After confirming sufficient bone regeneration in the defect area, the cover screw was removed and replaced with a healing abutment of an appropriate height. The surgical site was closed using absorbable sutures (4−0 Vicryl^®^, Ethicon, Somerville, NJ, USA), which were removed 7–10 days later.

### 2.7. Follow-Up Visit and Clinical Evaluation

Post-operative visits were conducted at one, three, and four months. Clinical photographs and periapical radiographs were taken to check on the soft tissue condition and adverse events. The pain intensity was measured using an 11-point numerical rating scale (NRS). The satisfaction with the procedure was assessed through a five-point scale questionnaire administered directly by the patients after the second-stage surgery.

### 2.8. Implant Stability and Dehiscence Defect Measurement

The implant stability was measured in the first- and second-stage surgery using an Ostell^®^ mentor device and Easy Check, similarly to the initial implant stability. A mucoperiosteal flap was elevated to allow for an exact assessment of the defect, and the clinical bone defect size was then measured in the same manner as the first-stage surgery (Figure 3). The defect size was measured using a periodontal probe (Colorvue Tip, Hu-Friedy, Leimen, Germany) with accompanying photographic documentation for all measurements. The photographs were analyzed using Image J 1.54g (NIH, Bethesda, MD, USA) software to measure the lengths. 

### 2.9. Statistical Analysis

The sample size for this clinical study was n = 14 for each group. Due to the limited sample size, the Shapiro–Wilk test was conducted to assess normal distribution. The results indicated that all findings had *p* > 0.05, suggesting non-normality of the data. In cases where data do not conform to a normal distribution, nonparametric tests are appropriate. Therefore, the Mann–Whitney test was used to evaluate differences in median values and the similarity of distributions between the two groups. To assess the association between each group and the patients’ gender and smoking status, a chi-square test was performed. A *p*-value of less than 0.05 was considered indicative of a significant difference between the groups. The level of statistical significance was set at *p* < 0.05, and statistical analysis was conducted using IBM SPSS 29.0 (IBM Corp., Armonk, NY, USA).

## 3. Results

### 3.1. Patient Demographic Information

This study conducted a suitability assessment on 28 patients who voluntarily signed informed consent forms to participate in the research. The demographic information of the study participants is presented in Table 2. The participants consisted of 14 males and 14 females. The SLA group had five males (35.71%) and nine females (64.29%), while the SLA/plasma group had nine males (64.29%) and five females (35.71%). The mean age of all participants was 62.82 ± 12.40 years, with the SLA group having a mean age of 64.21 ± 8.51 years and the SLA/plasma group having a mean age of 61.43 ± 15.58 years. Among the total participants, three individuals smoked, with two in the SLA group (14.29%) and one in the SLA/plasma group (7.14%). No statistically significant differences in the participants’ gender, age, or smoking status were observed between the two groups.

### 3.2. Implant Placement

The distribution of implants in the jaws among the participants showed 11 (39.29%) in the maxilla and 17 (60.71%) in the mandible. In the SLA group, 4 (28.57%) implants were placed in the maxilla and 10 (71.43%) in the mandible, while in the SLA/plasma group, seven (50.00%) implants were placed in both the maxilla and mandible. Regarding the distribution of implant locations, the SLA group and SLA/plasma group had five (35.71%) implants in the anterior region and nine (64.29%) in the posterior region (Table 3). 

### 3.3. Cumulative Survival Rate

Clinical examinations and radiographic evaluations were conducted to assess the safety of this study. The clinical examinations included assessing the signs of inflammation, bleeding, or suppuration on gentle probing, and the implant mobility was evaluated [26]. Therefore, the SLA and SLA/plasma groups showed 100% cumulative survival rates with no implant failure observed.

### 3.4. Clinical Examination of Buccal Bone Defect

#### 3.4.1. Measurement of Vertical Bone Defect

The vertical height of bone defects in the first-stage surgery was 1.37 ± 1.39 mm (range 0.00 to 4.76 mm) in the SLA group and 1.18 ± 1.09 mm (range 0.00 to 4.22 mm) in the SLA/plasma group, showing no statistical significance (*p* = 0.839). In the second-stage surgery, the defect height was 0.21 ± 0.44 mm (range 0.00 to 1.53 mm) in the SLA group and 0.29 ± 0.51 mm (range 0.00 to 1.72 mm) in the SLA/plasma group, showing no statistical significance (*p* = 0.667). The changes in defect height in both groups were 1.16 ± 1.58 mm for the SLA group and 0.89 ± 1.27 mm for the SLA/plasma group, indicating no significant difference in buccal bone defect height between the two groups (*p* = 0.541). The Shapiro–Wilk test indicated a lack of normality. Therefore, the Mann–Whitney test was used for analysis. Both groups showed a decrease in buccal bone defect height from the first-stage surgery to the second-stage surgery, indicating vertical bone regeneration in both groups (Figure 4, * *p* < 0.01, Mann–Whitney test), even though statistical significance between the groups was not observed (Figure 4 and Figure 5 and Table 4).

#### 3.4.2. Measurement of Horizontal Bone Defect

The horizontal depth of bone defects in the first-stage surgery was 0.85 ± 0.83 mm (range: 0.00 to 3.33 mm) in the SLA group and 0.87 ± 0.70 mm (range: 0.00 to 2.50 mm) in the SLA/plasma group, suggesting no statistical significance (*p* = 0.734). In the second-stage surgery, the defect depth was 0.18 ± 0.39 mm (range 0.00 to 1.39 mm) in the SLA group and 0.10 ± 0.28 mm (range 0.00 to 0.96 mm) in the SLA/plasma group, indicating no statistical significance (*p* = 0.571). The change in defect depth was 0.67 ± 1.02 mm and 0.77 ± 0.80 mm for the SLA and SLA/plasma groups, respectively, indicating no significant difference in buccal bone defect depth between the two groups (*p* = 0.734). Statistical analysis was conducted in the same manner for horizontal bone defect reconstruction as for vertical bone defect recovery. Similarly, while bone recovery over time was observed in each group (Figure 6, * *p* < 0.001, Mann−Whitney test), the results suggest that the extent of horizontal bone reconstruction was comparable between the two groups (Figure 6 and Figure 7 and Table 5).

### 3.5. Evaluation of Implant Stability

The stability of the implants was evaluated using ISQ values, measured with Ostell, and ISV values, measured with Easycheck. Measurements were taken at implant placement and during the second-stage surgery, with two measurements per implant averaged for an evaluation.

During the first-stage surgery, the ISQ values were 76.07 ± 4.22 (range: 68.50–85.00) for the SLA group and 76.04 ± 5.44 (range: 64.50–84.00) for the SLA/plasma group, showing no significant difference in initial implant stability between the two groups (*p* = 0.874). At the second-stage surgery, the ISQ values were 79.18 ± 4.87 (range: 70.00–88.50) for the SLA group and 76.73 ± 5.51 (range: 65.00–82.50) for the SLA/plasma group, showing no statistical significance (*p* = 0.402). The changes in ISQ were 3.11 ± 5.08 and 0.92 ± 7.61 for the SLA and SLA/plasma groups, respectively, with no significant difference between the two groups (*p* = 0.375) (Figure 8 and Figure 9 and Table 6).

Similarly to the ISQ measurement results, the ISV measurement results showed no significant difference between the groups during the first-stage surgery (SLA: 67.25 ± 4.81, SLA/plasma: 69.61 ± 9.66, *p* = 0.352) or the second-stage surgery (SLA: 76.89 ± 7.27, SLA/plasma: 75.29 ± 7.19, *p* = 0.603). The change in ISV values also showed no statistical significance (SLA: 9.64 ± 7.28, SLA/plasma: 5.68 ± 10.84, *p* = 0.454) (Figure 10 and Figure 11 and Table 7). 

These results confirmed the implant stability in the SLA and SLA/plasma groups.

### 3.6. Evaluation of Marginal Bone Change

The marginal bone changes were evaluated from periapical radiographs taken during the first-stage surgery, one and three months post-surgery, and during the second-stage surgery (four months after the first-stage surgery). Using the ImageJ program, mesial and distal marginal bone length were measured relative to the length of the implant fixture on the radiographs. The measurements were taken with reference to the implant platform, where values were recorded as positive above the platform and negative below it. The marginal bone changes were calculated by subtracting the marginal bone length measured at the time of the first-stage surgery from that measured at one and three months post-surgery and at the time of the second-stage surgery. 

Mesial side bone changes at one month showed significant differences between the SLA group (−0.32 ± 0.42 mm, range: −1.20–0.24 mm) and the SLA/plasma group (0.38 ± 0.63 mm, range: −0.39–2.14 mm) (*p* = 0.001). This trend continued at three months (SLA: −0.54 ± 0.49 mm, SLA/plasma: 0.84 ± 1.48 mm, *p* = 0.001) and at the second-stage surgery (SLA: −0.97 ± 1.82 mm, SLA/plasma: 1.15 ± 1.50 mm, *p* = 0.001), with the SLA/plasma group consistently showing significantly higher values (Figure 12, Table 8).

Similarly, the distal side bone changes exhibited significant differences between the groups at one month (SLA: −0.33 ± 0.59 mm, SLA/plasma: 0.39 ± 0.54 mm, *p* = 0.002), three months (SLA: −0.50 ± 0.64 mm, SLA/plasma: 0.46 ± 0.55 mm, *p* < 0.001), and second-stage surgery (SLA: −0.82 ± 2.32 mm, SLA/plasma: 0.58 ± 0.77 mm, *p* = 0.005). On the other hand, the changes over time within each group were similar (*p* > 0.05) (Figure 13, Table 9).

These findings suggest that marginal bone reconstruction was enhanced in the SLA/plasma group compared to the SLA group.

### 3.7. NRS (Numeric Rating Score) Pain Assessment 

When assessing patient pain on a scale of 0−10 divided into 11 steps, no statistically significant differences were observed between the SLA group (1 month: 0.57 ± 0.85, 3 months: 0.14 ± 0.53, second-stage surgery: 0.21 ± 0.43) and the SLA/plasma group (1 month: 0.29 ± 1.07, 3 months: 0.14 ± 0.36, second-stage surgery: 0.21 ± 0.80). These findings indicate similar outcomes in the pain levels in both groups at the specified time points (Table 10).

### 3.8. Participants’ Satisfaction with the Procedure

A patient satisfaction survey was conducted after all clinical trial procedures up to the second surgery. The results showed satisfaction scores of 4.21 ± 0.89 and 4.57 ± 0.51 for the SLA and SLA/plasma groups, respectively, with no significant difference between the two groups (*p* = 0.376) (Table 11). These findings suggest that both groups were generally satisfied with the clinical trial process.

## 4. Discussion

This prospective clinical study examined whether plasma treatment with the ACTILINK System can improve the surface characteristics of implants with SLA surfaces and determine if there is a clinical difference. The titanium metal used in dental implants has high corrosion resistance because the low solubility of its oxide layer prevents ion release when in contact with biological tissues [27]. The Ti−6Al−4V alloy, introduced to compensate for the mechanical limitation of pure titanium, is used widely as a biomaterial, including in dental implants, because of its high corrosion resistance and mechanical stability [28,29,30]. Various studies have been conducted to enhance osseointegration, and it has been revealed that the chemical composition, hydrophilicity, surface roughness, and geometric form of the implant surface affect cell adhesion [31]. Diverse surface treatment methods, such as sandblasting, acid etching, and anodization, have been applied to improve the surface characteristics of implant fixtures [32]. Nevertheless, aging occurs once the implant surface comes into contact with air, decreasing the hydrophilicity and biological activity [4,5,6]. As the aging process advances, hydrocarbon impurities develop on the implant surface, which has been demonstrated to impair the osseointegration performance of the implant [33]. Surface treatments involving UV and plasma have been developed to address this aging phenomenon [34]. Huang et al. compared implants treated with and without UVC in a beagle model experiment and showed higher bone–implant contact (BIC) and bone volume measured by CT in the treated group. On the other hand, no significant differences were observed based on different durations of UVC treatment (1/6, 1/2, and one hour) [10]. In a rabbit model experiment using dielectric barrier discharge plasma, a significant increase in bone–implant contact (BIC) was observed in the group treated with plasma. This effect was observed specifically in male rabbits, which was attributed to differences in osteoblast characteristics based on gender [8]. In another animal experiment involving plasma treatment, radiographic analysis revealed a significant reduction in implant crestal bone loss six weeks after plasma treatment compared to the control group [23]. 

This study examined the clinical differences based on vacuum plasma treatment in patients presenting with buccal bone defects of 3 mm or less during guided bone regeneration (GBR) at the time of implant placement. Vertical height measurements of buccal bone defects showed no clinical significance between the SLA and the SLA/plasma groups, but both groups showed bone healing over four months. Similarly, there were no significant differences in horizontal bone width changes between the two groups, but both showed evidence of bone healing. Previous animal experimental results, which involved sample harvesting six weeks post-plasma treatment, showed significant effects [8,23]. It was presumed that the implants in human subjects would require an adequate healing period to resist functional loads, which may not show significant clinical changes. Methods to evaluate implant stability include clinical measurements such as insertion torque resistance, reverse torque testing, and periostatological tests [35]. The ISQ measurement using Osstell indicates high stability when the values are above 70. 

In contrast, studies have shown that the ISQ values at implant placement do not predict prognosis, whereas the post-healing ISQ values after an adequate period may sufficiently prevent implant failure [36]. Implants placed in this study exhibited favorable outcomes, with mean ISQ values above 70 during secondary surgery, irrespective of the group. Easy Check is a method for evaluating implant stability by striking a healing abutment without additional equipment, which, like ISQ, indicates high stability with values above 70, according to the manufacturer. The ISV measurements also showed average values above 70 in both groups during secondary surgery. A histological examination for measuring bone-to-implant contact (BIC) is considered the gold standard for assessing osseointegration [37], despite the challenges of performing it in human studies. Therefore, in this study, the BIC was evaluated by measuring the distance from the platform to the marginal bone crest on the mesial and distal sides of the implant using periapical radiographs. As a result, the marginal bone changes were increased significantly in the SLA/plasma group compared to the SLA group on both the mesial and distal sides. Various studies have shown that plasma treatments can enhance the hydrophilicity of titanium alloys [38,39]. and improve cell attachment [40,41]. In vivo studies using rat and beagle models revealed increased bone formation around plasma-treated implants [21,42]. In particular, studies using miniature pig models observed increased interthread bone density [43]. 

Nevertheless, these studies primarily evaluated implants with all surfaces in contact with existing bone tissue. In the present study, while no significant differences in marginal bone changes between the SLA group and SLA/plasma group were observed on the buccal side with dehiscence, significant differences were observed on the mesial and distal sides where the implant was in complete contact with the existing bone tissue. Although no differences were observed between the two groups in terms of implant stability, such as ISQ and ISV, considering that marginal bone decreases over a certain period after function is applied to the implants [44,45], the higher bone healing observed in the SLA/plasma group at the mesial and distal sides is believed to positively impact the long-term prognosis of the implants. Therefore, plasma-treated implants may enhance implant stability, particularly in areas prone to dehiscence.

## 5. Conclusions

This prospective clinical study showed that plasma treatment with the ACTILINK System before SLA implant placement does not adversely affect the stability or efficacy of the implants. Significant increases in marginal bone were observed on the mesial and distal sides as measured by periapical radiographs, highlighting the need for further research on this aspect.

## Figures and Tables

**Figure 1 bioengineering-11-00980-f001:**
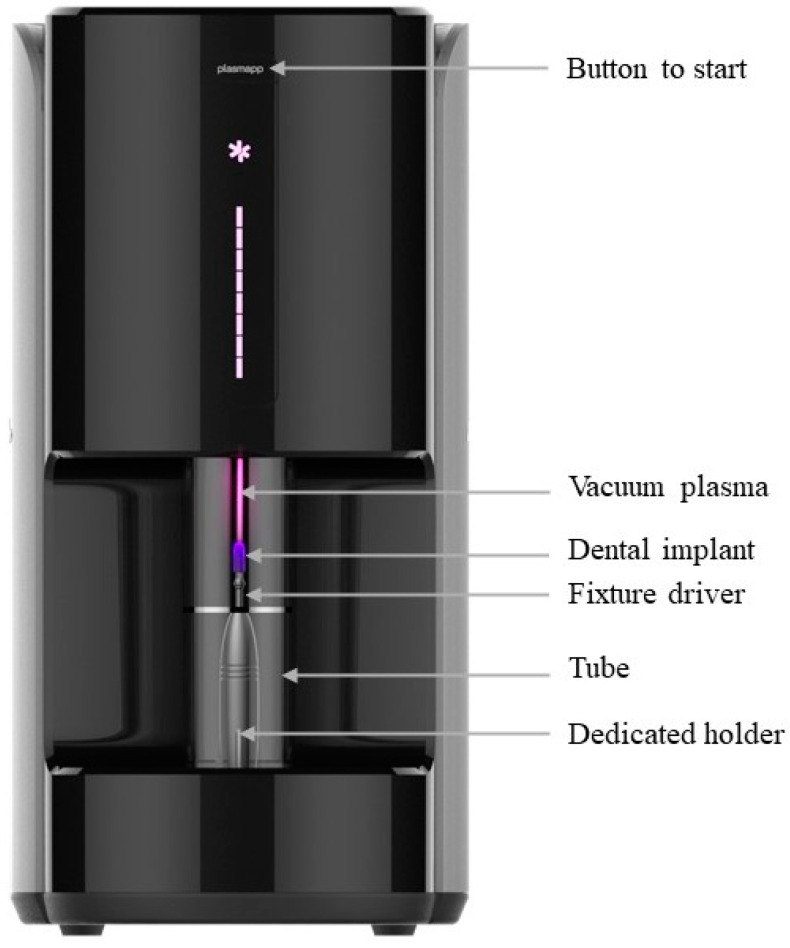
Appearance and structure of the ACTILINK system.

**Figure 2 bioengineering-11-00980-f002:**
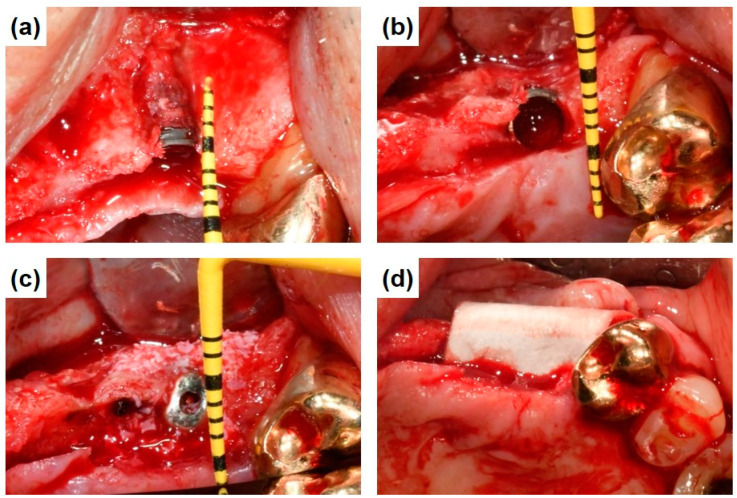
First-stage surgical procedures. (**a**) Peri-implant bone defect height, (**b**) peri-implant bone defect depth, (**c**) bone augmentation with BONE−D XB, and (**d**) application of Colla−DM.

**Figure 3 bioengineering-11-00980-f003:**
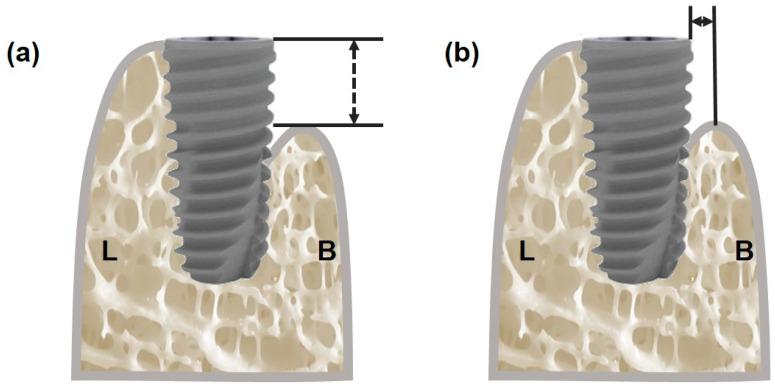
Clinical measurements of peri-implant defect size at implant surgery. (**a**) Defect height (mm) from the implant platform to the first bone-to-implant contact (BIC), and (**b**) defect depth (mm) from the bone crest to implant surface (in a direction perpendicular to implant’s major axis). (L: lingual, B: buccal).

**Figure 4 bioengineering-11-00980-f004:**
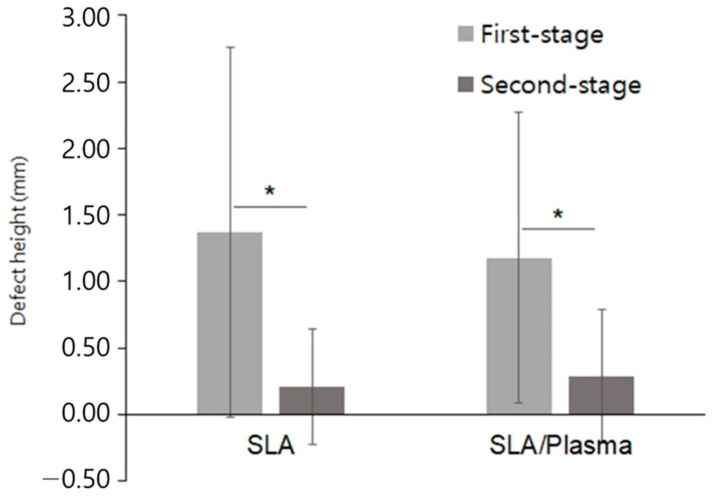
Vertical height of the buccal bone defects (* *p* < 0.01, Mann–Whitney test).

**Figure 5 bioengineering-11-00980-f005:**
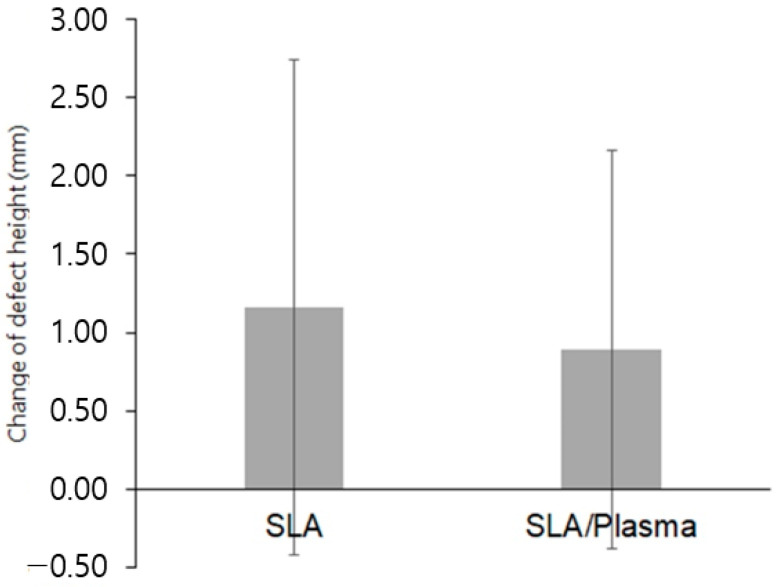
Change in the vertical height of the buccal bone defects.

**Figure 6 bioengineering-11-00980-f006:**
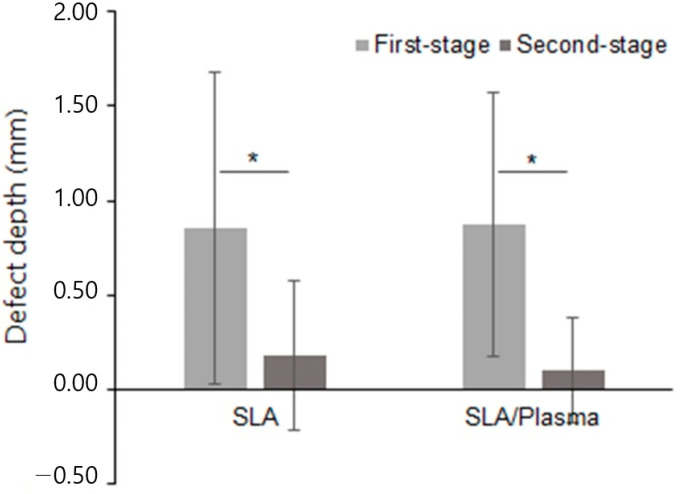
Horizontal depth of the buccal bone defects (* *p* < 0.001, Mann−Whitney test).

**Figure 7 bioengineering-11-00980-f007:**
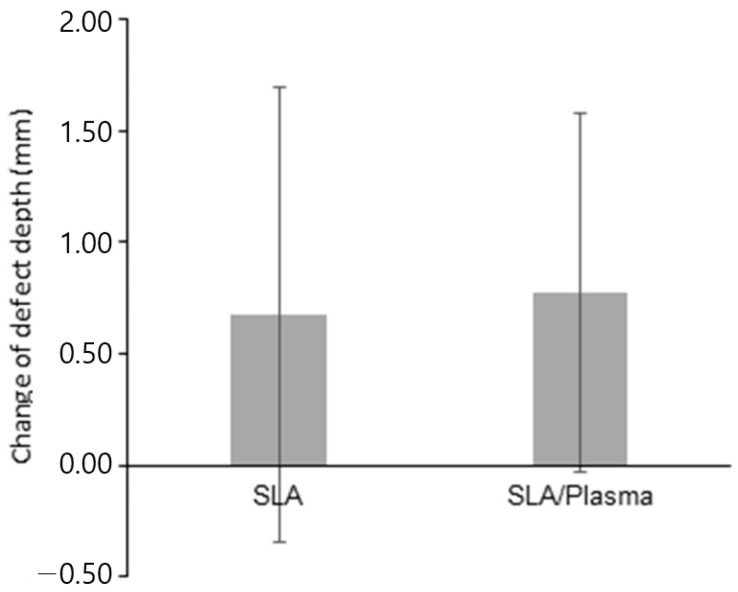
Change in the horizontal depth of the buccal bone defects.

**Figure 8 bioengineering-11-00980-f008:**
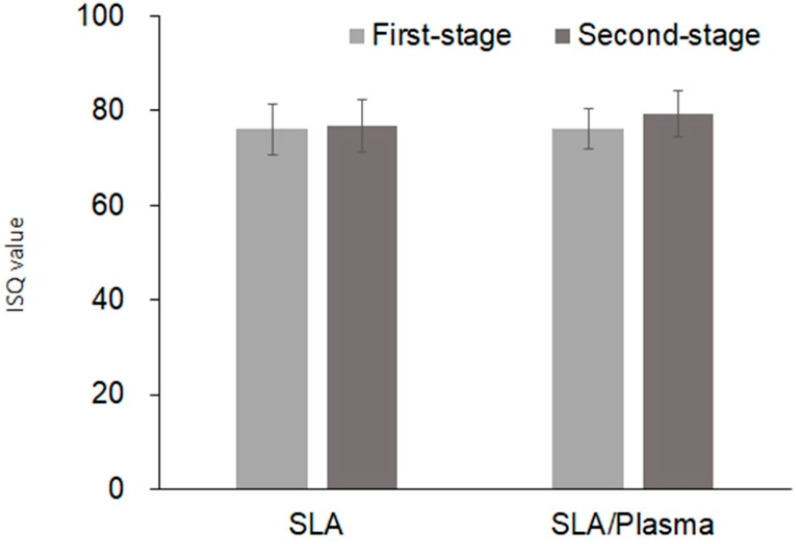
ISQ values at the first-stage and second-stage surgery.

**Figure 9 bioengineering-11-00980-f009:**
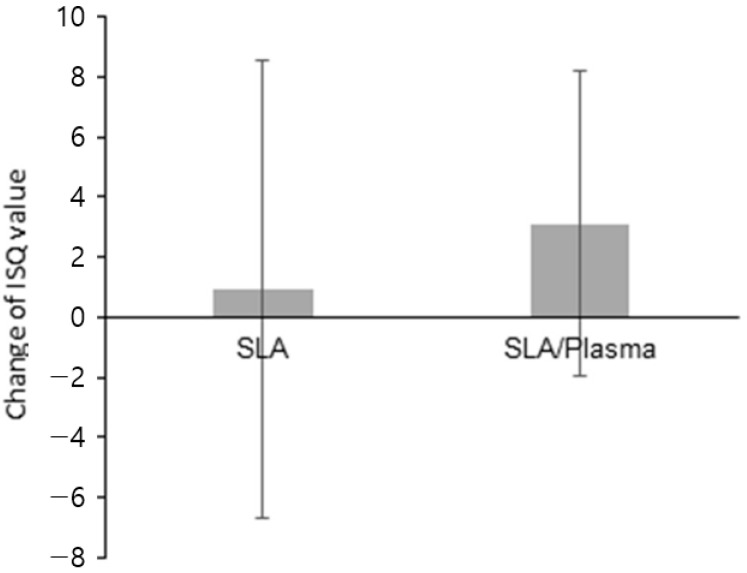
Difference in the ISQ values between the first-stage and second-stage surgery.

**Figure 10 bioengineering-11-00980-f010:**
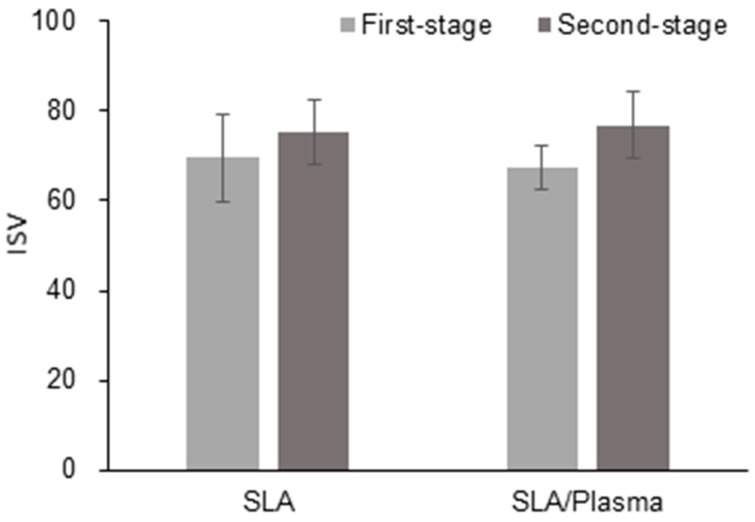
ISV at the first-stage and second-stage surgery.

**Figure 11 bioengineering-11-00980-f011:**
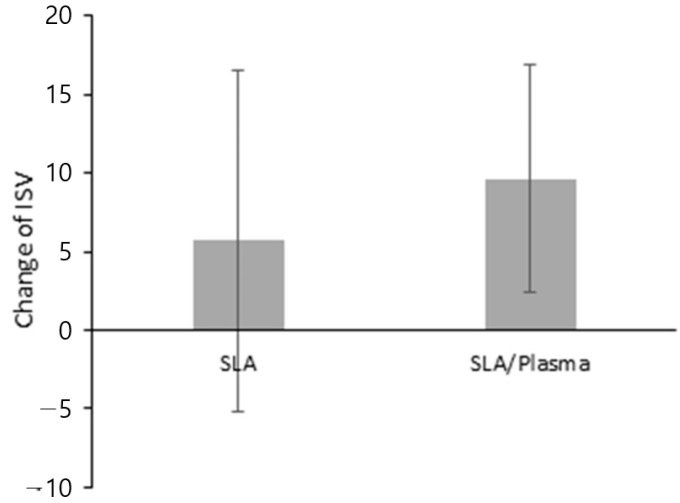
Difference in ISV between the first-stage and second-stage surgery.

**Figure 12 bioengineering-11-00980-f012:**
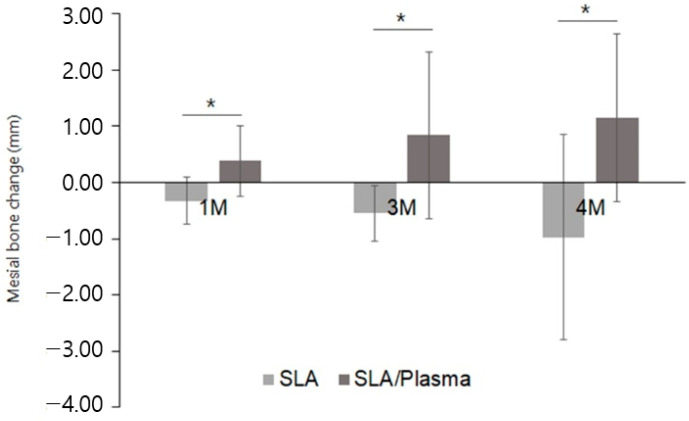
Marginal bone change in the mesial side (* *p* < 0.001, Mann–Whitney test).

**Figure 13 bioengineering-11-00980-f013:**
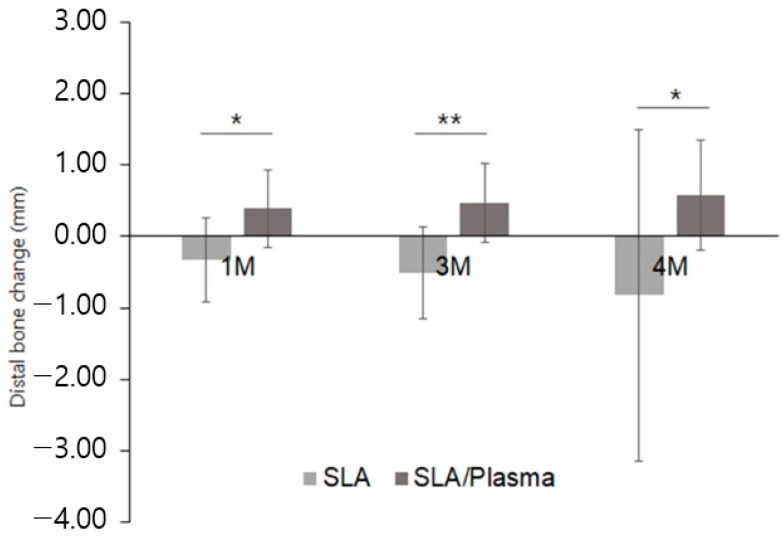
Marginal bone change in the distal side (* *p* < 0.01, ** *p* < 0.001, Mann–Whitney test).

**Table 1 bioengineering-11-00980-t001:** (a) Inclusion and (b) exclusion criteria of this study.

**(a) Inclusion Criteria**
(1) Patient aged 19 years or older (2) More than one missing tooth in either maxilla or mandible (3) Tooth missing or extraction before the last six weeks or more ago (4) Successful initial fixation of the implanted osseous dental implant (no mobility of the implant after surgery) (5) Peri-implant buccal dehiscence defect height under 3 mm (a simple autogenous bone graft is deemed necessary) (6) Adequate oral hygiene (Full Mouth Plaque Index (FMPI) and Full Mouth Bleeding on Probing (FMBoP) lower than or equal to 25%) (7) Willingness and ability to follow the study requirements and voluntary signature to the informed consent form
**(b) Exclusion Criteria**
(1) Heavy smoking (more than 20 cigarettes per day) (2) Uncontrolled metabolic disease (e.g., diabetes, osteomalacia, thyroid disease) (3) Patient at risk from the dental treatment or the surgical procedure (4) History of hypersensitivity to titanium (5) History of chemotherapy or radiation treatment during the last five-year period (6) Pregnant patient or patient with a pregnancy plan during the clinical trial (7) Patient in breastfeeding (8) Current treatment of metabolic bone disorders (e.g., osteoporosis) (9) History of corticosteroid therapy within the last 6-month period (cumulative dosage of 150 mg or more) (10) Presence of acute or chronic oral inflammation (e.g., osteomyelitis) (11) Current immunocompromised condition (12) Presence of oral mucosal disease or oral lesions (13) Current abnormal blood circulation (14) History of drug abuse or alcoholism (15) Poor oral hygiene status (16) Current participation in other clinical trials (17) Previous graft (bone augmentation) at the intended implant surgical site (18) Individuals with a subordinate relationship to the principal investigator, such as students, residents, or staff members (19) Other individuals deemed unsuitable for participation in this clinical trial

**Table 2 bioengineering-11-00980-t002:** Patients’ demographic information.

	Number of Patients (%)			*p*-Value
	SLA Group (n = 14)	SLA/Plasma Group (n = 14)	Total (n = 28)	
Age				
Mean ± SD	64.21 ± 8.51	61.43 ± 15.58	62.82 ± 12.40	0.839 ^a^
Median [Min–Max]	66.50 [51.00–82.00]	67.50 [23.00–78.00]	67.00 [23.00–82.00]	
Gender				
Male	5 (35.71)	9 (64.29)	14 (50.00)	0.131 ^b^
Female	9 (64.29)	5 (35.71)	14 (50.00)	
Smoking				
Smoker	2 (14.29)	1 (7.14)	3 (10.71)	0.541 ^b^
Non-smoker	12 (85.71)	13 (92.86)	25 (89.29)	

^a^ Mann–Whitney test; ^b^ Chi-square test.

**Table 3 bioengineering-11-00980-t003:** Implant placement.

	Number of Patients (%)		
	SLA Group (n = 14)	SLA/Plasma Group (n = 14)	Total (n = 28)
Location			
Maxilla	4 (28.57)	7 (50.00)	11 (39.29)
Mandible	10 (71.43)	7 (50.00)	17 (60.71)
Position			
Anterior	5 (35.71)	5 (35.71)	10 (35.71)
Posterior	9 (64.29)	9 (64.29)	18 (64.29)

**Table 4 bioengineering-11-00980-t004:** Measurement of the vertical height of the buccal bone defects.

	SLA Group (n = 14)	SLA/Plasma Group (n = 14)	*p*-Value ^a^
First stage			
Mean ± SD	1.37 ± 1.39	1.18 ± 1.09	0.839
Median [Min–Max]	1.09 [0.00–4.76]	1.00 [0.00–4.22]	
Second stage			
Mean ± SD	0.21 ± 0.44	0.29 ± 0.51	0.667
Median [Min–Max]	0.00 [0.00–1.53]	0.00 [0.00–1.72]	
Change in defect height ^b^			
Mean ± SD	1.16 ± 1.58	0.89 ± 1.27	0.541
Median [Min–Max]	1.89 [−1.21–4.76]	0.67 [−1.02–4.22]	

^a^ Mann–Whitney test; ^b^ (defect height at first-stage surgery)—(defect height at second-stage surgery).

**Table 5 bioengineering-11-00980-t005:** Measurement of the horizontal depth of the buccal bone defects.

	SLA Group (n = 14)	SLA/Plasma Group (n = 14)	*p*-Value ^a^
First-stage			
Mean ± SD	0.85 ± 0.83	0.87 ± 0.70	0.734
Median [Min–Max]	0.61 [0.00–3.33]	0.58 [0.00–2.50]	
Second-stage			
Mean ± SD	0.18 ± 0.39	0.10 ± 0.28	0.571
Median [Min–Max]	0.00 [0.00–1.39]	0.00 [0.00–0.96]	
Change in defect depth ^b^			
Mean ± SD	0.67 ± 1.02	0.77 ± 0.80	0.734
Median [Min–Max]	0.54 [−1.03–3.33]	0.55 [−0.36–2.50]	

^a^ Mann−Whitney test; ^b^ (defect depth at first-stage surgery)—(defect depth at second-stage surgery).

**Table 6 bioengineering-11-00980-t006:** Measurement of the implant stability using the ISQ value.

	SLA Group (n = 14)	SLA/Plasma Group (n = 14)	*p*-Value ^a^
First stage			
Mean ± SD	76.07 ± 4.22	76.04 ± 5.44	0.874
Median [Min–Max]	75.25 [68.50–85.00]	76.50 [64.50–84.00]	
Second stage			
Mean ± SD	79.18 ± 4.87	76.73 ± 5.51	0.402
Median [Min–Max]	79.00 [70.00–88.50]	78.50 [65.00–82.50]	
Change of ISQ value ^b^			
Mean ± SD	3.11 ± 5.08	0.92 ± 7.61	0.375
Median [Min–Max]	3.75 [−5.50–10.50]	1.50 [−10.00–16.50]	

^a^ Mann−Whitney test; ^b^ (ISQ value at second-stage surgery)—(ISQ value at first-stage surgery).

**Table 7 bioengineering-11-00980-t007:** Measurement of implant stability using ISV.

	SLA Group (n = 14)	SLA/Plasma Group (n = 14)	*p*-Value ^a^
First stage			
Mean ± SD	67.25 ± 4.81	69.61 ± 9.66	0.352
Median [Min–Max]	67.75 [59.00–77.50]	69.50 [49.50–85.00]	
Second stage			
Mean ± SD	76.89 ± 7.27	75.29 ± 7.19	0.603
Median [Min–Max]	79.00 [60.00–85.50]	75.00 [64.50–86.50]	
Change in ISV ^b^			
Mean ± SD	9.64 ± 7.28	5.68 ± 10.84	0.454
Median [Min–Max]	12.00 [−3.00–22.50]	6.25 [−10.50–20.00]	

^a^ Mann−Whitney test; ^b^ (ISV at second-stage surgery)—(ISV at first-stage surgery).

**Table 8 bioengineering-11-00980-t008:** Measurement of the mesial side bone changes.

	SLA Group (n = 14)	SLA/Plasma Group (n = 14)	*p*-Value ^d^
One month ^a^			
Mean ± SD	−0.32 ± 0.42	0.38 ± 0.63	0.001
Median [Min–Max]	−0.29 [−1.20–0.24]	0.28 [−0.39–2.14]	
Three months ^b^			
Mean ± SD	−0.54 ± 0.49	0.84 ± 1.48	0.001
Median [Min–Max]	−0.48 [−4.54–0.25]	0.51 [−0.54–5.20]	
Second-stage surgery (four months) ^c^			
Mean ± SD	−0.97 ± 1.82	1.15 ± 1.50	0.001
Median [Min–Max]	−0.55 [−6.90–0.05]	0.65 [0.00–5.60]	

^a^ (Mesial bone height at one month post-surgery)—(mesial bone height at first-stage surgery); ^b^ (mesial bone height at three months post-surgery)—(mesial bone height at first-stage surgery); ^c^ (mesial bone height at second-stage surgery)—(mesial bone height at first-stage surgery); ^d^ Mann–Whitney test.

**Table 9 bioengineering-11-00980-t009:** Measurement of the distal-side bone changes.

	SLA Group (n = 14)	SLA/Plasma Group (n = 14)	*p*-Value ^d^
One month ^a^			
Mean ± SD	−0.33 ± 0.59	0.39 ± 0.54	0.002
Median [Min–Max]	−0.45 [−1.34–0.80]	0.27 [−0.41–1.23]	
Three months ^b^			
Mean ± SD	−0.50 ± 0.64	0.46 ± 0.55	<0.001
Median [Min–Max]	−0.50 [−1.21–0.96]	0.39 [−0.30–1.36]	
Second-stage surgery (four months) ^c^			
Mean ± SD	−0.82 ± 2.32	0.58 ± 0.77	0.005
Median [Min–Max]	−0.48 [−8.10–1.36]	0.37 [−0.53–1.90]	

^a^ (Distal bone height at one month post-surgery)—(distal bone height at first-stage surgery); ^b^ (distal bone height at three months post-surgery)—(distal bone height at first-stage surgery); ^c^ (distal bone height at second-stage surgery)—(distal bone height at first-stage surgery); ^d^ Mann−Whitney test.

**Table 10 bioengineering-11-00980-t010:** NRS score at one and three months post-surgery, and at second-stage surgery.

	SLA Group (n = 14)	SLA/Plasma Group (n = 14)	*p*-Value ^a^
One month			
Mean ± SD	0.57 ± 0.85	0.29 ± 1.07	0.246
Median [Min–Max]	0.00 [0.00–2.00]	0.00 [0.00–4.00]	
Three months			
Mean ± SD	0.14 ± 0.53	0.14 ± 0.36	0.804
Median [Min–Max]	0.00 [0.00–1.00]	0.00 [0.00–1.00]	
Second-stage surgery			
Mean ± SD	0.21 ± 0.43	0.21 ± 0.80	0.571
Median [Min–Max]	0.00 [0.00–1.00]	0.00 [0.00–3.00]	

^a^ Mann−Whitney test.

**Table 11 bioengineering-11-00980-t011:** Participants’ satisfaction survey results.

	SLA Group (n = 14)	SLA/Plasma Group (n = 14)	*p*-Value ^a^
Mean ± SD	4.21 ± 0.89	4.57 ± 0.51	0.376
Median [Min–Max]	4.00 [2.00–5.00]	5.00 [4.00–5.00]	

^a^ Mann−Whitney test.

## Data Availability

All data has been presented in the manuscript.

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
