# Peer review of "Prospective Randomized Controlled Clinical Trial to Evaluate the Safety and Efficacy of ACTLINK Plasma Treatment for Promoting Osseointegration and Bone Regeneration in Dental Implants"

_bioengineering, 2024, doi:10.3390/bioengineering11100980_

Round 1
Reviewer 1 Report
Comments and Suggestions for Authors
Dear authors,
The bioengineering manuscript-321502, entitled 'Prospective Randomized Controlled Clinical Trial to Evaluate the Safety and Efficacy of ACTLINK Plasma Treatment for Promoting Osseointegration and Bone Regeneration in Dental Implants' presents the potential of pre-implantation plasma treatment to enhance bone regeneration around implants.
Moreover, the authors intend to present a clinical trial in order to evaluate the safety and efficacy of plasma treatment for promoting osseointegration and bone regeneration in dental implants.
The protocol for this medial trial is well established by the authors, having a cohort of 28 patients, divided into 2 groups, one group whose implants are considered controls, and one with implants subjected to plasma treatments.
The processes of osseointegration of the implants studied in the designated patients were followed by investigative methods specific to dentistry.
The obtained results show that the plasma treatments of the implants do not affect their stability and efficiency. Moreover, an increase in the marginal bone near the implant was observed in the case of plasma-treated implants, which suggests a further research session.
Overall, the text of the manuscript is carefully written and suitable for the theme of the journal. The figures inserted in the manuscript are of good quality and are described in the text. The conclusions are short but succinct.
However, as it appears from the title of the manuscript, more information about the plasma source (ACTILINK System, Plasmapp Co. Ltd., Daejeon, Korea) in terms of average plasma power, working gas, excited species in the plasma, duration treatment, etc., which could help the reader to understand why such treatment is important and how he can help in such situations. Moreover, a diagnosis of the plasma source, both electrical and optical, would also help, or at least a reference to this information. In the manuscript, there is not even a basic diagram of how the plasma treatment took place, or a photograph. What is being treated more precisely: the implant itself or the 'holder' that is fixed in the jaw bone... Such information must appear explicitly in the manuscript.
After adding this valuable and necessary information, I propose that this revised manuscript be considered for publication in the Bioengineering journal.
Author Response
Comment 1:
However, as it appears from the title of the manuscript, more information about the plasma source (ACTILINK System, Plasmapp Co. Ltd., Daejeon, Korea) in terms of average plasma power, working gas, excited species in the plasma, duration treatment, etc., which could help the reader to understand why such treatment is important and how he can help in such situations. Moreover, a diagnosis of the plasma source, both electrical and optical, would also help, or at least a reference to this information. In the manuscript, there is not even a basic diagram of how the plasma treatment took place, or a photograph. What is being treated more precisely: the implant itself or the 'holder' that is fixed in the jaw bone... Such information must appear explicitly in the manuscript.
Answer:
Detailed principles and methods for plasma treatment of dental implant surface using ACTILINK system have been added to section 2.2 and are as follows.
2.2 Plasma treatment of the dental implant.
Plasma treatment for dental implants was performed using ACTILINK system (Plasmapp Co. Ltd., Daejeon, Korea). ACTILINK system is a device developed to reduce surface impurities of the dental implant by vacuum plasma treatment (Figure 1). For plasma treatment, the dental implant is connected to the fixture driver. The fixture driver used here is one recommended by the manufacturer of the dental implant. The fixture driver with dental implant is mounted on a dedicated holder for ACTILINK system. When the prepared holder is mounted on ACTILINK system, the implant is electrically grounded to the device. The set vacuum plasma process is performed by pressing the button. When the process begins, the tube of ACTILINK system is lowered to the base and the inflow of external air is blocked. After that, a vacuum of less than 10 torr is formed inside the tube through pumping, and a voltage of approximately 3kV is supplied to the implant from the power electrode at the top of the tube to discharge plasma on the surface of the implant. Impurities on the implant surface are removed through plasma treatment, and after plasma treatment, impurities are removed through additional pumping using a vacuum pump. This process takes about 1 minute.
Reviewer 2 Report
Comments and Suggestions for Authors
Review Bioengineering-3215021
I have read the paper "Prospective Randomized Controlled Clinical Trial to Evaluate the Safety and Efficacy of ACTLINK Plasma Treatment for Promoting Osseointegration and Bone Regeneration in Dental Implants” which has been submitted to Journal Bioengineering. The manuscript shows the results about a prospective clinical study aimed to assess the clinical stability and efficacy of plasma treatment applied to implants with sand-blast−acid etching (SLA) surfaces before placement. Overall, it was written with clarity and precision, but needs some corrections and modifications, which are suggested below:
Line 46: “hydrophilicity of implant fixtures”: Hydrophilicity and osseointegration are not necessarily directly related. It is better to state that a wettable surface facilitates the primary processes of osseointegration.
From line 69 to 70: Please inform the parameters of the plasma used. Voltage, frequency, working pressure, atmosphere, distance and application time...
From line 99 to 102: Knowing that the aging time of a plasma-treated surface is crucial to the wettability value, it is important to inform the time used between the treatment and the insertion of the implant.
Table 2: Since osseointegration is associated with hormonal, vitamin and mineral deficiencies, it is very strange that the SLA control has more women than the SLA treated with plasma. This needs to be clearly justified.
Table 3: Differences in osseointegration for different locations, such as mandible and maxilla, historically proven, must be justified because it was preferentially used for one of the conditions studied.
Author Response
Comment 1: Line 46: “hydrophilicity of implant fixtures”: Hydrophilicity and osseointegration are not necessarily directly related. It is better to state that a wettable surface facilitates the primary processes of osseointegration.
Respond 1: As your suggestion I revised that sentence from “ Therefore, research on methods, such as UV [7] or plasma treatment [8], to enhance the hydrophilicity of implant fixtures, promote the interaction with bone, stimulate osteoblasts, and remove carbohydrate contamination from the implant surface is actively pursued.” to “Therefore, research on methods, such as UV [7] or plasma treatment [8], to enhance the hydrophilicity of implant fixtures, and remove carbohydrate contamination from the implant surface, thereby promoting the primary processes of osseointegration.”
Comment 2: From line 69 to 70: Please inform the parameters of the plasma used. Voltage, frequency, working pressure, atmosphere, distance and application time...
Respond 2:
Detailed principles and methods for plasma treatment of dental implant surface using ACTILINK system have been added to section 2.2 and are as follows.
2.2 Plasma treatment of the dental implant.
Plasma treatment for dental implants was performed using ACTILINK system (Plasmapp Co. Ltd., Daejeon, Korea). ACTILINK system is a device developed to reduce surface impurities of the dental implant by vacuum plasma treatment (Figure 1). For plasma treatment, the dental implant is connected to the fixture driver. The fixture driver used here is one recommended by the manufacturer of the dental implant. The fixture driver with dental implant is mounted on a dedicated holder for ACTILINK system. When the prepared holder is mounted on ACTILINK system, the implant is electrically grounded to the device. The set vacuum plasma process is performed by pressing the button. When the process begins, the tube of ACTILINK system is lowered to the base and the inflow of external air is blocked. After that, a vacuum of less than 10 torr is formed inside the tube through pumping, and a voltage of approximately 3kV is supplied to the implant from the power electrode at the top of the tube to discharge plasma on the surface of the implant. Impurities on the implant surface are removed through plasma treatment, and after plasma treatment, impurities are removed through additional pumping using a vacuum pump. This process takes about 1 minute.
Comment 3: From line 99 to 102: Knowing that the aging time of a plasma-treated surface is crucial to the wettability value, it is important to inform the time used between the treatment and the insertion of the implant.
Respond 3:
The plasma-treated implant fixtures were implanted in patients immediately after the treatment. To clarify, we have added the following information to Materials and Methods section 2.4.
For the SLA/plasma group, dental implant fixtures underwent plasma treatment using the ACTILINK System device before implant insertion. The plasma-treated implant fixtures were inserted in patients immediately after the treatment.
Comment 4: Table 2: Since osseointegration is associated with hormonal, vitamin and mineral deficiencies, it is very strange that the SLA control has more women than the SLA treated with plasma. This needs to be clearly justified.
Respond4: When assigning participants to experimental groups, we used a randomized program to allocate them to either the SLA group or the SLA/plasma group based on the order of participation, without giving special consideration to gender. In future clinical studies, we will take your suggestion into account and consider gender in the study design. Thank you for your valuable feedback.
Comment 5: Table 3: Differences in osseointegration for different locations, such as mandible and maxilla, historically proven, must be justified because it was preferentially used for one of the conditions studied.
Respond: As with the patient's gender, the experimental groups were randomly assigned without considering the implant site. We believe that accounting for factors like gender or implant location may impose significant limitations on the study. However, it would be beneficial to consider these factors in future research. Thank you for your valuable feedback.
Reviewer 3 Report
Comments and Suggestions for Authors
Comments to the manuscript bioengineering-3215021: Prospective Randomized Controlled Clinical Trial to Evaluate the Safety and Efficacy of ACTLINK Plamsa Treatment for Promoting Osseoinegration and Bone Regeneration in Dental Implants
The present manuscript deals with the possibility of an NTP treatment of dental implants and the time after implementation of these implants observing healing procedure. The paper is interesting and underlines similar studies using plasma systems. English is fine. The topic fits to the chosen journal. However, there are some minor aspects which should be addressed to further improve the quality of this manuscript. They are as followed:
· Line 53: The wide field of plasma applications should be concisely introduced, i.e. applications in surface etching and environmental application in abatement of contaminants or desinfection should be mentioned beside medical application. Adequate literature references, which might be very helpful, are as followed::
o https://doi.org/10.1016/j.jece.2018.03.012
o https://doi.org/10.1016/j.chemosphere.2023.138061
o https://doi.org/10.1016/j.msec.2021.112474
o https://doi.org/10.1142/S0219581X23300080
Perhaps it should be mentioned that NTP process efficiency may be increased by combination with catalysts or even biological processes, i.e. microorganisms may show high resistance against plasma treatment requiring high specific energy levels or optimized treatment conditions. Maybe helpful references are:
o https://doi.org/10.3390/su12208577
o https://doi.org/10.1007/s41614-022-00077-1
· Line 55: It would be helpful for non-expert readers to understand reaction of the plasma, which cause enhanced grade of hydrophilicity, i.e. please list the physicochemical effects of reduced surface tension (https://doi.org/10.1016/j.polymertesting.2021.107097) as well as immobilization of amines or carboxy groups at the surface (https://doi.org/10.1016/j.nano.2016.05.014).
· Line 150: Both statistic tests (Shapiro-Wilk, Mann-Whitney) should be shortly explained according application, derived results, how to perform the statistic, pros and cons of the test.
Author Response
Comments 1: Line 53: The wide field of plasma applications should be concisely introduced, i.e. applications in surface etching and environmental application in abatement of contaminants or desinfection should be mentioned beside medical application. Adequate literature references, which might be very helpful, are as followed::
o https://doi.org/10.1016/j.jece.2018.03.012
o https://doi.org/10.1016/j.chemosphere.2023.138061
o https://doi.org/10.1016/j.msec.2021.112474
o https://doi.org/10.1142/S0219581X23300080
Respond 1: Thank you for your helpful advice. I have revised the introduction based on the articles you mentioned, and the updated content is as follows:
Plasma is an ionized gas in which electrons and ions move freely, and it could be used in various fields [11-14]. It is typically formed when high temperature or electrical energy are applied, consisting of ions, electrons, neutral particles, and various form of ra-diation, such as UV. In a literature review study on surface modification using plasma, it was noted that plasma can be effectively used to create antibacterial coatings, and ongoing research aims to induce antibacterial and osteoinductive properties in metallic biomateri-als through plasma treatment. Additionally, it was stated that plasma has the potential to meet biocompatibility, cost-effectiveness, reproducibility, and industrial productivity re-quirements for surface treatment of biomaterials [15]. In another review article on surface treatment using plasma, it was reported that plasma exhibits properties that enhance surface adhesion, corrosion resistance, hardness, and wear resistance when applied to metal surfaces [16]. Additionally, plasma can enhance the biocompatibility of the surface while retaining the favorable bulk material properties of biomaterials, which plays a cru-cial role in their integration with the biological environment [17].
Comments 2:
Perhaps it should be mentioned that NTP process efficiency may be increased by combination with catalysts or even biological processes, i.e. microorganisms may show high resistance against plasma treatment requiring high specific energy levels or optimized treatment conditions. Maybe helpful references are:
o https://doi.org/10.3390/su12208577
o https://doi.org/10.1007/s41614-022-00077-1
Respond 2: Thank you for your valuable feedback. I believe there are some differences in the direction of the paper, so I have added examples of various fields where plasma can be utilized in the introduction.
Comments 3: Line 55: It would be helpful for non-expert readers to understand reaction of the plasma, which cause enhanced grade of hydrophilicity, i.e. please list the physicochemical effects of reduced surface tension (https://doi.org/10.1016/j.polymertesting.2021.107097) as well as immobilization of amines or carboxy groups at the surface (https://doi.org/10.1016/j.nano.2016.05.014).
Respond 3: Thank you for your kind review. The first paper discusses plasma treatment of polymers, while the second paper focuses on tissue engineering using electrospun scaffolds. It seems that these topics differ somewhat from the direction of this paper. I will keep them in mind for future related research.
Comments 4: Line 150: Both statistic tests (Shapiro-Wilk, Mann-Whitney) should be shortly explained according application, derived results, how to perform the statistic, pros and cons of the test.
Respond 4:
Detailed information about statistics test have been added to section 2.9 and are as follow:
2.9 Statistical analysis
The sample size for this clinical study was n=14 for each group. Due to the limited sample size, the Shapiro-Wilk test was conducted to assess normal distribution. The results indicated that all findings had p > 0.05, suggesting non-normality of the data. In cases where data do not conform to a normal distribution, nonparametric tests are appropriate. Therefore, the Mann-Whitney test was used to evaluate differences in median values and the similarity of distributions between the two groups. To assess the association between each group and the patients' gender and smoking status, a chi-square test was performed. A p-value of less than 0.05 was considered indicative of a significant difference between the groups. The level of statistical significance was set at p<0.05, and the statistical analysis was conducted using IBM SPSS 29.0 (IBM Corp., Armonk, NY, USA).
Reviewer 4 Report
Comments and Suggestions for Authors
The manuscript focuses on plasma treatment as a surface modification technique to promote osseointegration, which is a relevant and emerging area of study in dental implantology. Plasma treatment has been discussed in various studies, but the combination with SLA-treated implants in a clinical setting is an original contribution. This study addresses the gap in clinical evidence for plasma-treated implants and its potential to improve clinical outcomes over traditional SLA treatment alone.
Compared to other studies, which focus on in-vitro or animal model research on plasma-treated implants, this manuscript adds clinical trial data, providing direct evidence in human subjects. It highlights potential advantages like increased hydrophilicity and bone-to-implant contact, which have been supported by experimental studies in the literature.
-Improvements: The manuscript lacks specific details regarding the randomization process, which should be explained more clearly to ensure transparency. Additionally, the inclusion of a power analysis would strengthen the study's design and demonstrate the adequacy of the sample size.
- Further Controls:The study could benefit from including a broader set of controls. For instance, a UV-treated implant group could provide a better comparison of different hydrophilic surface treatments. Furthermore, long-term follow-up data beyond four months would offer insights into the sustained effects of plasma treatment.
The conclusions claim that plasma treatment significantly increases marginal bone levels on the mesial and distal sides of implants. This is supported by the statistical analysis presented in the manuscript, where significant differences were observed in bone changes between the groups. However, the lack of significant differences in implant stability (measured by ISQ) and the small sample size should be discussed more thoroughly to avoid overstating the conclusions. Additionally, the clinical relevance of these differences in marginal bone levels needs to be explained in more detail.
The references are generally appropriate, citing relevant studies about plasma treatment, osseointegration, and surface modification of implants. However, some more recent studies exploring longer-term outcomes of plasma treatment or comparisons with other hydrophilic treatments (e.g., UV treatment) could be included to enhance the depth of the literature review.
The tables are generally clear and provide adequate information. However, the text does not always refer to the tables in an integrated way, which can cause some disconnection between the results and their interpretation.
Figure 3, which illustrates the Kaplan-Meier survival curve, does not add significant information to the study, as both groups show a 100% survival rate. This figure could be removed without losing any valuable content.
The quality of the data is generally sound, but the statistical analysis could be expanded. The authors should discuss the potential impact of confounding variables (e.g., smoking or other health conditions) in greater depth.
Comments:
- Introduction: The introduction is too brief and does not provide sufficient background on the current state of the field. It would benefit from a more comprehensive review of plasma treatment in implantology and its comparison with other surface treatment methods like UV activation.
- Discussion:The discussion is focused, but more attention could be given to the implications of the findings for clinical practice, especially in terms of the cost-benefit analysis of plasma treatment versus conventional treatments.
- Minor Issues: The authors should improve the flow between the sections to ensure that all results are connected to the hypothesis and broader implications of the research. Additionally, grammatical errors and inconsistent formatting need to be addressed.
-Methodological clarity:The randomization and blinding processes need to be described in more detail.
-Power analysis: Including a power analysis to ensure the study is not underpowered.
-References:The manuscript should include more recent studies that directly compare plasma and UV treatments.
-Clarity: Ensure that the figures and tables are well integrated into the narrative of the manuscript.
In conclusion, the manuscript addresses an important clinical question regarding the efficacy of plasma treatment in dental implants. While the results are promising, the authors should address the methodological concerns and provide a more in-depth analysis of the findings to strengthen their argument.
Author Response
Comments 1: Improvements: The manuscript lacks specific details regarding the randomization process, which should be explained more clearly to ensure transparency. Additionally, the inclusion of a power analysis would strengthen the study's design and demonstrate the adequacy of the sample size.
Respond 1:
The criteria for determining the sample size have been added to Materials & Methods section 2.1, as follows:
2.1. Randomized controlled clinical trial
This study was a prospective, single−blinded, non−inferiority, randomized, and controlled clinical trial to evaluate the safety and efficacy of vacuum plasma treat-ment using the ACTILINK System (Plasmapp Co. Ltd., Daejeon, Korea). According to the most recent study by Jung et al. evaluating the efficacy of bone regeneration sur-gery using implants, the new bone regeneration rate was the highest, with a mean ± standard deviation of 96.4 ± 10 (%). We assume the lowest mean value from previous studies, 86% from Hämmerle et al., as the "acceptable maximum value" for satisfying the non-inferiority of this study. For sample size calculation, a one-sided significance level of 0.025 and a power of 80% were assumed. The number of subjects needed to satisfy the non-inferiority hypothesis was calculated using PASS 13 (NCSS Statistics Software, Kaysville, UT), resulting in 14 subjects per group, for a total of 28 subjects in-cluded in this clinical study. This prospective study protocol was approved by the In-stitutional Review Board of Pusan National University Dental Hospital (IRB Approval No. 2023−05−020).
2.4. Treatment group allocation
Twenty−eight participants were randomly assigned into two groups using the Mi-crosoft Excel 2016 (Microsoft Corp., Redmond, WA, USA). A list of participants was entered, and random numbers were generated to allocate participants to groups ran-domly based on these numbers.
- SLA group; non−plasma treated SLA implants, n = 14.
• SLA/plasma group; plasma treated SLA implants, n = 14.
Comments 2: Further Controls:The study could benefit from including a broader set of controls. For instance, a UV-treated implant group could provide a better comparison of different hydrophilic surface treatments. Furthermore, long-term follow-up data beyond four months would offer insights into the sustained effects of plasma treatment.
Respond 2: In future studies, we will conduct comparative research on the effects of UV and plasma, along with long-term observations. Thank you for your valuable suggestions.
Comments 3: The conclusions claim that plasma treatment significantly increases marginal bone levels on the mesial and distal sides of implants. This is supported by the statistical analysis presented in the manuscript, where significant differences were observed in bone changes between the groups. However, the lack of significant differences in implant stability (measured by ISQ) and the small sample size should be discussed more thoroughly to avoid overstating the conclusions. Additionally, the clinical relevance of these differences in marginal bone levels needs to be explained in more detail.
Respond 3:
Following your advice, I have added relevant content to the discussion along with references, and the details are as follows:
Although no differences were observed between the two groups in terms of implant stability, such as ISQ and ISV, considering that marginal bone decreases over a certain period after function is applied to the implants [44,45] the higher bone healing observed in the SLA/plasma group at the mesial and distal sides is believed to positively impact the long-term prognosis of the implants.
Comments 4: The references are generally appropriate, citing relevant studies about plasma treatment, osseointegration, and surface modification of implants. However, some more recent studies exploring longer-term outcomes of plasma treatment or comparisons with other hydrophilic treatments (e.g., UV treatment) could be included to enhance the depth of the literature review
Respond 4:
Although there are no papers that directly compare UV and plasma treatments, I have included introductions and references related to each, as detailed below:
In a study comparing the effects of UV−A and UV−C treatment on titanium implant surface for 40 minutes, the results showed that the contact angle of untreated implants was over 90o, which decreased to below 5o after UV−A treatment and to 34o after UV−C treatment [7]. A study comparing cell adhesion and mRNA expression in vitro after a UV and plasma treatment reported that a 12−minute UV treatment and plasma treatment for one minute resulted in the highest cell growth [9]. In addition, research findings have suggested no significant differences in implant stability and marginal bone loss after a UV−C treatment in vivo [10].
Plasma is an ionized gas in which electrons and ions move freely, and it could be used in various fields [11-14]. It is typically formed when high temperature or electrical energy are applied, consisting of ions, electrons, neutral particles, and various form of ra-diation, such as UV. In a literature review study on surface modification using plasma, it was noted that plasma can be effectively used to create antibacterial coatings, and ongoing research aims to induce antibacterial and osteoinductive properties in metallic biomateri-als through plasma treatment. Additionally, it was stated that plasma has the potential to meet biocompatibility, cost-effectiveness, reproducibility, and industrial productivity re-quirements for surface treatment of biomaterials [15]. In another review article on surface treatment using plasma, it was reported that plasma exhibits properties that enhance surface adhesion, corrosion resistance, hardness, and wear resistance when applied to metal surfaces [16]. Additionally, plasma can enhance the biocompatibility of the surface while retaining the favorable bulk material properties of biomaterials, which plays a cru-cial role in their integration with the biological environment [17].
Plasma can be categorized into thermal and non-thermal types [17], and non-thermal plasma can be generated at room temperature and has the advantage of short processing times.
Comments 5: The tables are generally clear and provide adequate information. However, the text does not always refer to the tables in an integrated way, which can cause some disconnection between the results and their interpretation.
Respond 5: I have reviewed the text to ensure that it is cohesively connected with the tables.
Comments 6: Figure 3, which illustrates the Kaplan-Meier survival curve, does not add significant information to the study, as both groups show a 100% survival rate. This figure could be removed without losing any valuable content.
Respond 6: As per your feedback, the cumulative survival rate graph has been excluded from the results.
Comments 7: The quality of the data is generally sound, but the statistical analysis could be expanded. The authors should discuss the potential impact of confounding variables (e.g., smoking or other health conditions) in greater depth.
Respond:
Detailed information about statistics test have been added to section 2.9 and are as follow:
2.9 Statistical analysis
The sample size for this clinical study was n=14 for each group. Due to the limited sample size, the Shapiro-Wilk test was conducted to assess normal distribution. The results indicated that all findings had p > 0.05, suggesting non-normality of the data. In cases where data do not conform to a normal distribution, nonparametric tests are appropriate. Therefore, the Mann-Whitney test was used to evaluate differences in median values and the similarity of distributions between the two groups. To assess the association between each group and the patients' gender and smoking status, a chi-square test was performed. A p-value of less than 0.05 was considered indicative of a significant difference between the groups. The level of statistical significance was set at p<0.05, and the statistical analysis was conducted using IBM SPSS 29.0 (IBM Corp., Armonk, NY, USA).
Comments 8: - Introduction: The introduction is too brief and does not provide sufficient background on the current state of the field. It would benefit from a more comprehensive review of plasma treatment in implantology and its comparison with other surface treatment methods like UV activation.
Respond 8:
Although there are no papers that directly compare UV and plasma treatments, I have included introductions and references related to each, as detailed below:
In a study comparing the effects of UV−A and UV−C treatment on titanium implant surface for 40 minutes, the results showed that the contact angle of untreated implants was over 90o, which decreased to below 5o after UV−A treatment and to 34o after UV−C treatment [7]. A study comparing cell adhesion and mRNA expression in vitro after a UV and plasma treatment reported that a 12−minute UV treatment and plasma treatment for one minute resulted in the highest cell growth [9]. In addition, research findings have suggested no significant differences in implant stability and marginal bone loss after a UV−C treatment in vivo [10].
Plasma is an ionized gas in which electrons and ions move freely, and it could be used in various fields [11-14]. It is typically formed when high temperature or electrical energy are applied, consisting of ions, electrons, neutral particles, and various form of ra-diation, such as UV. In a literature review study on surface modification using plasma, it was noted that plasma can be effectively used to create antibacterial coatings, and ongoing research aims to induce antibacterial and osteoinductive properties in metallic biomateri-als through plasma treatment. Additionally, it was stated that plasma has the potential to meet biocompatibility, cost-effectiveness, reproducibility, and industrial productivity re-quirements for surface treatment of biomaterials [15]. In another review article on surface treatment using plasma, it was reported that plasma exhibits properties that enhance surface adhesion, corrosion resistance, hardness, and wear resistance when applied to metal surfaces [16]. Additionally, plasma can enhance the biocompatibility of the surface while retaining the favorable bulk material properties of biomaterials, which plays a cru-cial role in their integration with the biological environment [17].
Plasma can be categorized into thermal and non-thermal types [17], and non-thermal plasma can be generated at room temperature and has the advantage of short processing times.
Comments 9: Discussion: The discussion is focused, but more attention could be given to the implications of the findings for clinical practice, especially in terms of the cost-benefit analysis of plasma treatment versus conventional treatments.
Respond 9:
I have added a discussion on the potential impact of this study on actual clinical practice, detailed below:
Although no differences were observed between the two groups in terms of implant stability, such as ISQ and ISV, considering that marginal bone decreases over a certain period after function is applied to the implants [44,45] the higher bone healing observed in the SLA/plasma group at the mesial and distal sides is believed to positively impact the long-term prognosis of the implants.
Comments 10: - Minor Issues: The authors should improve the flow between the sections to ensure that all results are connected to the hypothesis and broader implications of the research. Additionally, grammatical errors and inconsistent formatting need to be addressed.
Respond 10: I have reviewed the text to ensure that it is cohesively connected with the tables.
Comments 11: -Methodological clarity:The randomization and blinding processes need to be described in more detail.
Respond 11:
The details regarding the random assignment of participants have been added to Materials and Methods section 2.4, as follows:
2.4. Treatment group allocation
Twenty−eight participants were randomly assigned into two groups using the Mi-crosoft Excel 2016 (Microsoft Corp., Redmond, WA, USA). A list of participants was entered, and random numbers were generated to allocate participants to groups ran-domly based on these numbers.
- SLA group; non−plasma treated SLA implants, n = 14.
• SLA/plasma group; plasma treated SLA implants, n = 14.
Comments 12: -Power analysis: Including a power analysis to ensure the study is not underpowered.
Respond 12:
The criteria for determining the sample size have been added to Materials & Methods section 2.1, as follows:
2.1. Randomized controlled clinical trial
This study was a prospective, single−blinded, non−inferiority, randomized, and controlled clinical trial to evaluate the safety and efficacy of vacuum plasma treat-ment using the ACTILINK System (Plasmapp Co. Ltd., Daejeon, Korea). According to the most recent study by Jung et al. evaluating the efficacy of bone regeneration sur-gery using implants, the new bone regeneration rate was the highest, with a mean ± standard deviation of 96.4 ± 10 (%). We assume the lowest mean value from previous studies, 86% from Hämmerle et al., as the "acceptable maximum value" for satisfying the non-inferiority of this study. For sample size calculation, a one-sided significance level of 0.025 and a power of 80% were assumed. The number of subjects needed to satisfy the non-inferiority hypothesis was calculated using PASS 13 (NCSS Statistics Software, Kaysville, UT), resulting in 14 subjects per group, for a total of 28 subjects in-cluded in this clinical study. This prospective study protocol was approved by the In-stitutional Review Board of Pusan National University Dental Hospital (IRB Approval No. 2023−05−020).
Comments 13: -References: The manuscript should include more recent studies that directly compare plasma and UV treatments.
Respond 13: Although there are no papers that directly compare UV and plasma treatments, I have included introductions and references related to each.
Comments 14: -Clarity: Ensure that the figures and tables are well integrated into the narrative of the manuscript.
Respond 14: I have reviewed the text to ensure that it is cohesively connected with the figures and tables.
Round 2
Reviewer 4 Report
Comments and Suggestions for Authors
Following the revisions made in response to the feedback provided, I am pleased to formally accept this manuscript for publication. The authors have adequately addressed the concerns raised, and the improvements made have enhanced the overall clarity and quality of the work. I am confident that the article is now suitable for publication in its current form